# Antioxidative effects of molybdenum and its association with reduced prevalence of hyperuricemia in the adult population

Joo Hong Joun[1‡], Lilin Li[2,3], Jung Nam An[4], Joonho Jang[1], Yun Kyu Oh [2,5], Chun Soo Lim[2,5], Yon Su Kim[2,6], Kyungho Choi[7], Jung Pyo Lee[2,5], Jeonghwan Lee [2,5]*

1 Seoul National University Hospital, Seoul, Korea, 2 Department of Internal Medicine, Seoul National University College of Medicine, Seoul, Korea, 3 Department of Intensive Care Unit, Yanbian University Hospital, Jilin, China, 4 Department of Internal Medicine, Hallym University Sacred Heart Hospital, Anyang, Korea, 5 Department of Internal Medicine, Seoul National University Boramae Medical Center, Seoul, Korea, 6 Department of Internal Medicine, Seoul National University Hospital, Seoul, Korea, 7 Department of Environmental Health Sciences, School of Public Health, Seoul National University, Seoul, Korea

‡ JHJ is the first author on this work.
* jeonghwan@snu.ac.kr

**Data Availability Statement:** NHANES study protocols were approved by the National Center for Health Statistics ethics review board (https://www.cdc.gov/nchs/nhanes/irba98.htm). Urinary metal

## Abstract

The relationship between molybdenum and kidney-related disease outcomes, including hyperuricemia, is not well investigated. This study aims to determine whether molybdenum and its antioxidative property are associated with systemic inflammation and kidney-related disease parameters including hyperuricemia. Urinary molybdenum's epidemiological relationship to hyperuricemia and kidney-disease related outcomes was evaluated in 15,370 adult participants in the National Health and Nutrition Examination Survey (NHANES) collected between 1999 and 2016. Individuals' urinary molybdenum levels were corrected to their urinary creatinine concentrations. The association between urinary molybdenum-to-creatinine ratio and kidney-disease related outcomes were assessed by multivariable linear and logistic regression analyses, adjusting for covariates including age, sex, ethnicity, diabetes mellitus, hypertension, body mass index, and estimated glomerular filtration rate. Antimony and tungsten were used as control trace metals. Experimentally, HK-2 cell was used to assess molybdenum's antioxidative properties. HK-2 cells were challenged with $H_2O_2$-induced oxidative stress. Oxidative stress was measured using a fluorescent microplate assay for reactive oxygen species (ROS) and antioxidation levels were assessed by measuring the expression of manganese superoxide dismutase. In the adult NHANES population, urinary molybdenum-to-creatinine ratio was significantly associated with decreased serum uric acid (β, -0.119; 95% CI, -0.148 to -0.090) concentrations, and decreased prevalence of hyperuricemia (OR, 0.73; 95% CI, 0.64–0.83) and gout (OR, 0.71; 95% CI, 0.52–0.94). Higher urinary molybdenum levels were associated with lower levels of systemic oxidative stress (gamma-glutamyltransferase levels; β, -0.052; 95% CI, -0.067 to -0.037) and inflammation (C-reactive protein levels; β, -0.184; 95% CI, -0.220 to -0.148). In HK-2 cells under $H_2O_2$-induced oxidative stress, molybdenum upregulated manganese superoxide dismutase expression and decreased oxidative stress. Urinary molybdenum levels are associated with decreased prevalence of hyperuricemia and gout in adult population.

concentrations, demographic, and laboratory data were obtained from the NHANES database (https://wwwn.cdc.gov/nchs/nhanes/Default.aspx) in March 2019.

**Funding:** Jeonghwan Lee received funding of this study. This work was supported by a multidisciplinary research grant-in-aid from the Seoul Metropolitan Government Seoul National University (SMG-SNU) Boramae Medical Center (04-2023-0040). The funders had no role in study design, data collection and analysis, decision to publish, or preparation of the manuscript.

**Competing interests:** The authors declare no conflict of interest.

Molybdenum's antioxidative properties might have acted as an important mechanism for the reduction of systemic inflammation, ROS, and uric acid levels.

## Introduction

Molybdenum is a transition metal with a crucial role in the metabolic processes of microorganisms, plants, and animals [1, 2]. Human metabolism of carbon, sulfur, and nitrogen each utilizes aldehyde, sulfite, and xanthine oxidase, respectively; all three depend on their cofactor molybdenum for oxidant capacity [3]. The dietary requirement for molybdenum is small; Recommended Dietary Allowance (RDA) is 45 μg/day and Tolerable Upper Intake Level (UL) is 2,000 μg/day [4]. Legumes, grains, and nuts, particularly those grown in molybdenum-based fertilizer or molybdenum-rich soil, have a high molybdenum content [5, 6]. Clinically, tetrathiomolybdate is used to treat Wilson's disease to chelate free and food copper [7, 8].

As the sizable gap between RDA and UL suggests, humans have a high tolerance for molybdenum; deficiency or toxicity is rare [9]. One report of deficiency linked it to low serum uric acid concentration and Parkinson's disease [10], while others have associated high dietary and respiratory molybdenum exposure to hyperuricemia and gout-like symptoms [11, 12]. These studies proposed high molybdenum intake promotes xanthine oxidase activity, whose products are uric acid and reactive oxygen species (ROS). These observations, however, are not comparable to others that have documented no relationship between molybdenum intake and serum uric acid concentration [13, 14]. Although the relationship between molybdenum and particular groups such as those on hemodialysis and diabetes and chronic kidney disease (CKD) patients has been investigated previously [15–17], a population-wide association between molybdenum and health remains unclear and underreported.

In this study, we aimed to understand the antioxidative potential of molybdenum and the association between urinary molybdenum and various kidney-related parameters and comorbidities including gout and hyperuricemia. For this purpose, a human kidney proximal tubular epithelial cell line, i.e., HK-2, and a database from the National Health and Nutrition Examination Survey (NHANES) between 1999 and 2016 were employed. We investigated the antioxidative effects of molybdenum on HK-2 cells as well as the association between urinary molybdenum concentrations and systemic inflammation and disease outcomes, mainly hyperuricemia.

## Methods

### Study participants

In this population-based cohort study, we analyzed data of 15,370 adults aged ≥ 18 years who participated in the NHANES between 1999 and 2016 with urinary molybdenum concentrations available. NHANES study protocols were approved by the National Center for Health Statistics ethics review board (https://www.cdc.gov/nchs/nhanes/irba98.htm). All participants signed informed consent before study enrolment. This cross-sectional observational cohort study was approved by the Institutional Review Board of Seoul National University Boramae Medical Center (07-2019-35) and conducted in accordance with the Declaration of Helsinki [18]. Urinary metal concentrations, demographic, and laboratory data were obtained from the NHANES database (https://www.cdc.gov/nchs/nhanes/index.html) in March 2019. Participants' personal information was not disclosed, so researchers were able to identify participants by their study serial number. Urinary metal concentrations were measured via inductively

coupled plasma mass spectroscopy. To adjust for individuals' urinary dilution differences, urinary metal concentrations were corrected with urinary creatinine concentrations. For contrast, antimony and tungsten were also measured and analyzed as negative control elements to molybdenum. Tungsten was chosen in particular as it is also an essential trace element not known to have antioxidative or toxic properties and biochemically similar to molybdenum, sometimes even interchangeable in enzymes [19, 20].

The estimated glomerular filtration rate (eGFR) was calculated using the Chronic Kidney Disease Epidemiology Collaboration (CKD-EPI) equations [21]. Hypertension was defined as a systolic blood pressure > 140 mmHg or diastolic blood pressure > 90 mmHg being measured twice or more, history of hypertension, or current use of anti-hypertensive medication. Diabetes mellitus was defined as fasting glucose concentration > 126 mg/dL, random glucose concentration > 200 mg/dL, or history of diabetes mellitus.

## Kidney disease-related parameters, outcomes, and systemic inflammation

In this study, we examined the following kidney disease-related parameters: blood urea nitrogen (BUN), serum creatinine, urinary albumin-to-creatinine ratio (ACR), eGFR, and serum uric acid. Hyperuricemia was defined as a serum uric acid concentration over 6.0 mg/dL for females and over 7.0 mg/dL for males [22, 23]. Gout was defined using patients' reported questionnaire about their medical conditions as having been diagnosed with gout by a doctor (only available from NHANES dataset 2007–2016). We utilized serum gamma-glutamyltransferase levels as an indicator for systemic oxidative stress, and ferritin and C-reactive protein levels as inflammation parameters.

## In vitro tests of antioxidative effects of molybdenum

HK-2 human kidney proximal tubule epithelial cell line (Korean Cell Line Bank, Korea) was utilized to determine whether molybdenum and its antioxidative property. Molybdenum (powder form, 1–5 μm, ≥ 99.9% trace metals basis; Sigma-Aldrich, USA) was diluted in triple-distilled water to 2 mg/mL at room temperature and filter-sterilized with a .45-μm syringe filter (Sartorius, Germany) to obtain extracts [24]. Extracts were then further diluted in serum-free media for treatment. Aqueous $H2O2$ (3 wt. % in H2O, 200 ppm acetanilide as a stabilizer; Sigma) was diluted in serum-free media for treatment. Detailed methods of *in vitro* tests are explained in the **S1 Method**.

To determine the effect of molybdenum on cell survival and proliferation, a thiazolyl blue tetrazolium bromide (MTT; Sigma) assay was tested. Following serum starvation, cells were treated up to 400 μg/mL of molybdenum for 24 h. Following MTT (0.5 mg/mL dH$_2$O) incubation for 4 h at 37˚C, dimethyl sulfoxide was added to dissolve the resultant formazan. To observe cellular oxidative stress and antioxidative defense at the protein level, western blotting for manganese superoxide dismutase (MnSOD) (1:500; Cell Signaling Technology, USA) was performed. Intracellular ROS was evaluated by a fluorescent microplate assay using the 2',7'–dichlorofluorescein diacetate (DCFDA) Cellular ROS Detection Assay Kit (Abcam, USA).

## Statistical analysis

Urinary metal concentrations were divided by urine creatinine concentrations and the resulting urinary metal-to-creatinine ratio was used for analysis. Because urinary metal-to-creatinine ratios showed a right-skewed distribution, they were log-transformed to show normal distribution. Multivariable linear regression analysis was performed to examine the association between log-transformed urinary molybdenum-to-creatinine ratio and urinary ACR (log-transformed), BUN, serum creatinine, and eGFR, adjusting for age, sex, ethnicity, diabetes

mellitus, hypertension, and body mass index (BMI); the model for serum uric acid was additionally adjusted for eGFR. Urinary molybdenum-to-creatinine ratio's association with the prevalence of hyperuricemia and gout were investigated by multivariable logistic regression methods, adjusted with the same aforementioned factors as for serum uric acid. In the sensitivity analysis, models were further adjusted using covariates of socioeconomic status, education status, and smoking. The family poverty-to-income ratio as a continuous variable was used for socioeconomic status adjustment. Subgroup analysis was performed according to sex, age group, and BMI criteria. We visualized the risk of hyperuricemia according to urinary molybdenum levels using cubic splines (R package 'rms'). Relationship between urinary molybdenum and kidney-disease, systemic oxidative stress and inflammation parameters were tested with multivariable linear regression methods. Variables those were not in normal distribution were all natural log transformed.

Independent-samples t-tests were used to compare between two groups, accounting for the normality of the data. We performed one-way ANOVA using Tukey's test when comparing more than two groups. Statistical significance was determined at $P < 0.05$. All statistical analyses were performed using SPSS version 24 (IBM software, USA) and R 4.0.4, from January 1, 2019, to December 26, 2021.

## Results

### Baseline characteristics of the human population

**Table 1** summarizes the baseline characteristics. The median and interquartile range for urinary molybdenum were 43.4 (22.7–73.7) ng/mL, and those for the molybdenum-to-creatinine ratio were 39.5 (27.0–58.6) ng/mg. Participants with higher urinary molybdenum-to-creatine levels had more than twice the urinary molybdenum-to-creatine ratio [58.6 (47.6, 79.1) vs. 27.0 (20.0, 33.3); $P < 0.001$]. Adults in the upper half of urinary molybdenum-to-creatinine ratios displayed significantly lower (mean ± standard deviation) body weight (77.8 ± 19.8 vs 82.6 ± 21.5; $P < 0.001$), BMI (28.2 ± 6.5 to 28.9 ± 6.8; $P < 0.001$), and serum albumin (4.2 ± 0.4 vs 4.3 ± 0.4; $P < 0.001$), creatinine (0.85 ± 0.28 vs 0.93 ± 0.39; $P < 0.001$), and uric acid (5.2 ± 1.4 vs 5.6 ± 1.5; $P < 0.001$) concentrations, and proportion of male sex (44.6% vs 53.9%; $P < 0.001$) compared to those in the lower half of urinary molybdenum-to-creatinine ratios; mean age (45.9 ± 19.6 vs 45.8 ± 18.7; $P < 0.001$), BUN (13.5 ± 5.9 vs 13.0 ± 5.6; $P < 0.001$), urinary ACR (median, 7.8 [IQR, 4.5–19.4] vs 6.8 [4.4–12.4]; $P = 0.002$), eGFR (95.4 ± 24.6 vs 91.7 ± 23.7; $P < 0.001$), and proportion of hypertension (39.2% vs 37.9%; $P = 0.091$) and diabetes mellitus (15.2% vs 11.2%; $P < 0.001$) were significantly higher in the upper half of urinary molybdenum-to-creatinine ratios than in the lower half. There were more hyperuricemic adults in the lower half of molybdenum-to-creatinine ratios (n = 1,694; 22.0%) than in the upper half (n = 1,205; 15.7%) ($P < 0.001$), for a total of 2,899 (18.9%) hyperuricemic adults.

### Urinary molybdenum concentration and kidney disease-related parameters

Urinary molybdenum concentrations were compared between eGFR levels. Urinary molybdenum-to-creatinine ratio increased between groups with higher eGFR (**S1 Table**). Higher urinary molybdenum-to-creatinine ratio was correlated with increased eGFR (β, 4.78; 95% CI, 4.402–5.157; $P < 0.001$) and decreased serum creatinine (β, -0.064; 95% CI, -0.072 to -0.057; $P < 0.001$) and uric acid concentrations (β, -0.119; 95% CI, -0.148 to -0.090; $P < 0.001$), the last being additionally adjusted for eGFR (**Table 2**). The association between urinary molybdenum and BUN concentration was insignificant, but ACR showed an upward trend as urinary molybdenum-to-creatinine ratio increased.

**Table 1. Baseline characteristics according to urinary molybdenum levels.**

| | All | Lower half of urinary molybdenum-to-creatinine levels | Higher half of urinary molybdenum-to-creatinine levels | P value |
|---|---|---|---|---|
| | (N = 15,370) | (N = 7,685) | (N = 7,685) | |
| Male sex (%) | 7,566 (49.2%) | 4,142 (53.9%) | 3,424 (44.6%) | < 0.001 |
| Age, mean (SD), years | 47.2 ± 19.2 | 45.8 ± 18.7 | 45.9 ± 19.6 | < 0.001 |
| Race (%) | | | | < 0.001 |
| Mexican American | 2,953 (19.2%) | 1,223 (15.9%) | 1,730 (22.5%) | |
| Other Hispanic | 1,257 (8.2%) | 547 (7.1%) | 710 (9.2%) | |
| Non-Hispanic White | 6,777 (44.1%) | 3,343 (43.5%) | 3,434 (44.7%) | |
| Non-Hispanic Black | 3,147 (20.5%) | 2,070 (26.9%) | 1,077 (14.0%) | |
| Another race | 1,236 (8.0%) | 502 (6.5%) | 734 (9.6%) | |
| Body weight, mean (SD), kg | 80.2 ± 20.8 | 82.6 ± 21.5 | 77.8 ± 19.8 | < 0.001 |
| BMI, mean (SD), kg/m$^2$ | 28.6 ± 6.7 | 28.9 ± 6.8 | 28.2 ± 6.5 | < 0.001 |
| Diabetes mellitus (%) | 2,027 (13.2%) | 857 (11.2%) | 1,170 (15.2%) | < 0.001 |
| Hypertension (%) | 5,920 (38.5%) | 2,909 (37.9%) | 3,011 (39.2%) | 0.091 |
| Hyperuricemia (%) | 2,899 (18.9%) | 1,694 (22.0%) | 1,205 (15.7%) | < 0.001 |
| Blood urea nitrogen, mean (SD), mg/dL | 13.3 ± 5.8 | 13.0 ± 5.6 | 13.5 ± 5.9 | < 0.001 |
| Serum albumin, mean (SD), g/dL | 4.2 ± 0.4 | 4.3 ± 0.4 | 4.2 ± 0.4 | < 0.001 |
| Serum creatinine, mean (SD), mg/dL | 0.89 ± 0.34 | 0.93 ± 0.39 | 0.85 ± 0.28 | < 0.001 |
| Serum uric acid, mean (SD), mg/dL | 5.4 ± 1.4 | 5.6 ± 1.5 | 5.2 ± 1.4 | < 0.001 |
| Urinary ACR, median (IQR), mg/g | 6.9(4.4–13.3) | 6.6(4.2–12.4) | 7.3(4.7–14.4) | 0.002 |
| eGFR, mean (SD), mL/min/1.73 m$^2$ | 93.6 ± 24.2 | 91.7 ± 23.7 | 95.4 ± 24.6 | < 0.001 |
| GGT, mean (SD), U/L | 28.8 ± 41.8 | 30.0 ± 42.2 | 27.6 ± 41.4 | < 0.001 |
| Ferritin, mean (SD), ng/mL | 92.4 ± 128.8 | 96.1 ± 128.7 | 88.9 ± 128.9 | 0.038 |
| C-reactive protein, mean (SD), mg/dL | 0.44 ± 0.78 | 0.48 ± 0.93 | 0.39 ± 0.62 | < 0.001 |
| Urinary molybdenum, mean (SD), ng/mL | 56.7 ± 53.8 | 36.0 ± 25.2 | 77.5 ± 65.5 | < 0.001 |
| , median (IQR), ng/mL | 43.4 (22.7, 73.7) | 30.6 (16.0, 49.8) | 63.2 (35.7, 99.2) | |
| Urinary molybdenum-to-creatinine ratio, mean (SD), ng/mg | 49.7 ± 41.2 | 26.1 ± 8.6 | 73.2 ± 46.9 | < 0.001 |
| , median (IQR), ng/mg | 39.5 (27.0, 58.6) | 27.0 (20.0, 33.3) | 58.6 (47.6, 79.1) | |

Abbreviations: ACR, albumin-to-creatinine ratio; BMI, body mass index; eGFR, estimated glomerular filtration rate; GGT, gamma-glutamyltransferase.

Hyperuricemia is defined as a serum uric acid concentration of over 6.0 mg/dL for females and over 7.0 mg/dL for males.

SI Conversion factors: To convert urea nitrogen to millimoles per liter, multiply by 0.357; creatinine to micromoles per liter, multiply by 88.4; uric acid to millimoles per liter, multiply by 0.0595; glomerular filtration to mL/s/m$^2$, multiply by 0.0167.

**Table 2. Association between urinary molybdenum levels[a] and various kidney disease-related parameters.**

|  | Univariable analysis | | | Multivariable analysis | |
|---|---|---|---|---|---|
|  | β (95% CI) | P value |  | β (95% CI) | P value |
| Urinary ACR[a], mg/g | 0.112 (0.085, 0.140) | < 0.001 |  | 0.029 (0.003, 0.055) | 0.030 |
| eGFR, mL/min/1.73 m$^2$ | 2.813 (2.229, 3.396) | < 0.001 |  | 4.780 (4.402, 5.157) | < 0.001 |
| Blood urea nitrogen, mg/dL | 0.372 (0.234, 0.511) | < 0.001 |  | 0.074 (-0.051, 0.199) | 0.244 |
| Serum creatinine, mg/dL | -0.076 (-0.084, -0.068) | < 0.001 |  | -0.064 (-0.072, -0.057) | < 0.001 |
| Serum uric acid[b], mg/dL | -0.359 (-0.393, -0.324) | < 0.001 |  | -0.119(-0.148, -0.090) | < 0.001 |

Abbreviations: ACR, albumin-to-creatinine ratio; eGFR, estimated glomerular filtration rate.

[a]Urinary molybdenum-to-creatinine ratio (ng/mg) and urinary ACR were natural log-transformed.

Multivariable linear regression analyses were adjusted for age, sex, ethnicity, BMI, diabetes mellitus, and hypertension.

[b]Multivariable linear regression for serum uric acid was adjusted for age, sex, ethnicity, BMI, diabetes mellitus, hypertension, and estimated glomerular filtration rate.

### Risk of hyperuricemia according to urinary molybdenum concentrations

Urinary molybdenum-to-creatinine ratio was negatively correlated with the prevalence of hyperuricemia after adjusting for age, sex, ethnicity, BMI, diabetes mellitus, hypertension, and eGFR (Table 3); the association was consistent and significant throughout all second (OR, 0.87; 95% CI, 0.77–0.98; $P$ = 0.022), third (OR, 0.78; 95% CI, 0.69–0.88; $P$ < 0.001), and fourth (OR, 0.73; 95% CI, 0.64–0.83; $P$ < 0.001) quartiles (Q1-Q4) of urinary molybdenum-to-creatinine ratio ($P$ < 0.001 for trend) (Fig 1). However, neither antimony (Q2 OR, 0.99; 95% CI, 0.88–1.12; $P$ = 0.92; Q3 OR, 1.03; 95% CI, 0.91–1.17; $P$ = 0.63; Q4 OR, 1.06; 95% CI, 0.93–1.21; $P$ = 0.37; $P$ = 0.75 for trend) nor tungsten (Q2 OR, 1.08; 95% CI, 0.96–1.22; $P$ = 0.21; Q3 OR, 1.01; 95% CI, 0.89–1.14; $P$ = 0.89; Q4 OR, 0.97; 95% CI, 0.85–1.10; $P$ = 0.63; $P$ = 0.37 for trend) was significantly correlated at any quartile levels with the risk of hyperuricemia (Table 3).

In the sensitivity analysis, multivariable logistic regression models were further adjusted using covariates of socioeconomic status, education levels, and smoking status (S2 Table). The effects of urinary molybdenum to hyperuricemia were stratified according to sex, age group, and BMI criteria (S3–S5 Tables and Fig 2). Uric acid lowering effect of molybdenum more

**Table 3. Association between urinary metal levels and prevalence of hyperuricemia.**

|  | Molybdenum | | | Antimony | | Tungsten | |
|---|---|---|---|---|---|---|---|
|  | OR (95% CI) | P value |  |  | P value | OR (95% CI) | P value |
| Urinary metal-to-creatinine ratio |  | < 0.001 |  |  | 0.749 |  | 0.366 |
|  |  | < 0.001[*] |  |  | 0.317[*] |  | 0.461[*] |
| Q1, reference | 1 |  |  | 1 |  | 1 |  |
| Q2 | 0.87 (0.77, 0.98) | 0.022 |  | 0.99 (0.88–1.12) | 0.919 | 1.08 (0.96, 1.22) | 0.206 |
| Q3 | 0.78 (0.69, 0.88) | < 0.001 |  | 1.03 (0.91–1.17) | 0.633 | 1.01 (0.89, 1.14) | 0.886 |
| Q4 | 0.73 (0.64, 0.83) | < 0.001 |  | 1.06 (0.93–1.21) | 0.373 | 0.97 (0.85, 1.10) | 0.631 |

Abbreviation: Q1–Q4, quartile group of urinary metal levels.

Urinary molybdenum-to-creatinine ratio, median (IQR), ng/mg, Q1 20.0 (15.1, 23.7), Q2 33.3 (30.3, 36.4), Q3 47.6 (43.4, 52.7), Q4 79.1 (66.5, 106,4)

Urinary antimony-to-creatinine ratio, median (IQR), ng/mg, Q1 0.03 (0.02, 0.04), Q2 0.05 (0.05, 0.06), Q3 0.08 (0.07, 0.09), Q4 0.15 (0.13, 0.22)

Urinary tungsten-to-creatinine ratio, median (IQR), ng/mg, Q1 0.03 (0.02, 0.03), Q2 0.05 (0.05, 0.06), Q3 0.09 (0.08, 0.10), Q4 0.19 (0.14, 0.28)

[*]$P$-for-trend

Hyperuricemia is defined as a serum uric acid concentration of over 6.0 mg/dL for females and over 7.0 mg/dL for males.

Multivariate logistic regression analysis was adjusted for age, sex, ethnicity, BMI, diabetes mellitus, hypertension, and estimated glomerular filtration rate.

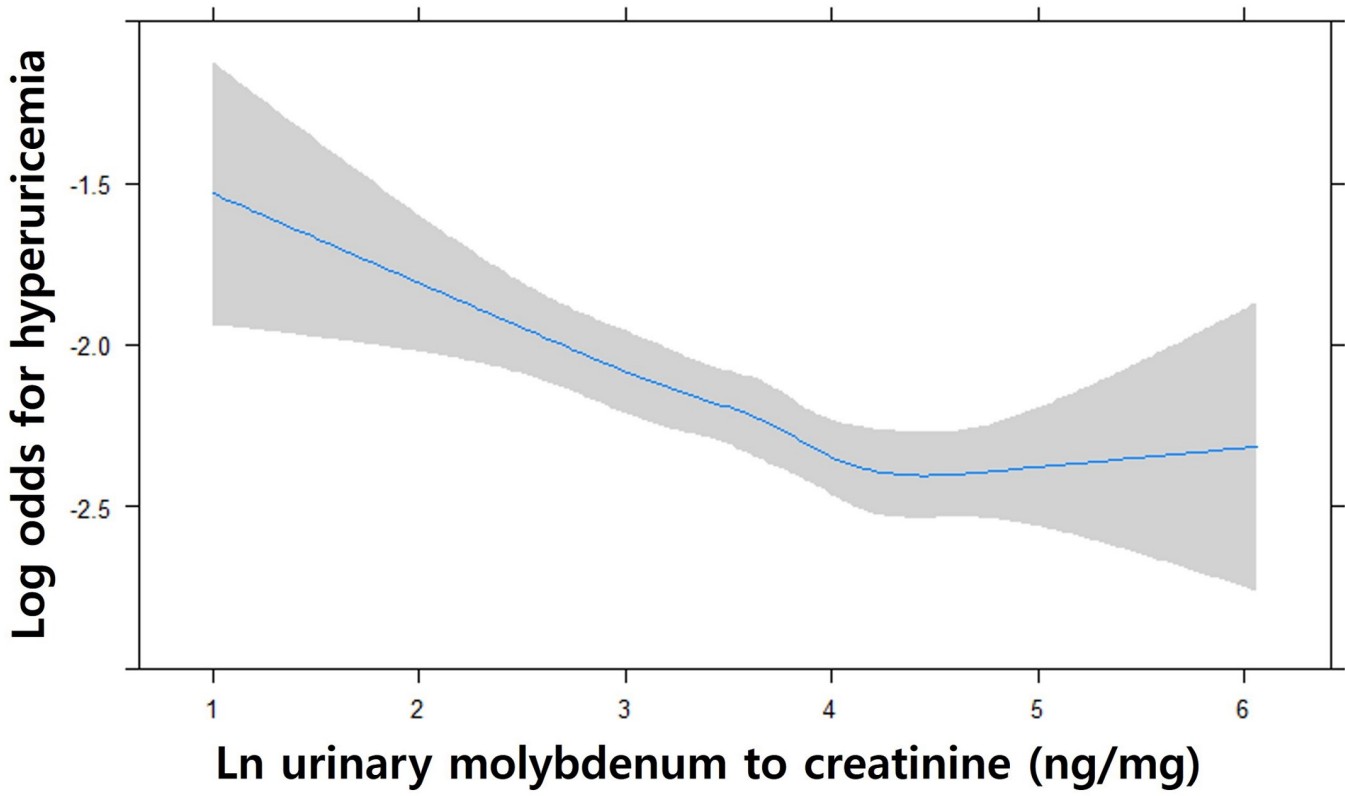

**Fig 1. Adjusted cubic splines for hyperuricemia according to log-transformed urinary molybdenum-to-creatinine levels.** The log value of the odds ratio for hyperuricemia decreased according to the increase of urinary molybdenum-to-creatinine levels. Covariates with age, sex, ethnicity, diabetes mellitus, hypertension, body mass index (BMI), and eGFR were considered in the adjusted model.

pronounced in men, younger participants, and overweight participants, and less significant in elderly patients over 65 years old and normal weight participants.

### Risk of gout according to urinary molybdenum concentrations

Table 4 summarizes the relationship between prevalence of gout and urinary molybdenum-to-creatinine ratio. Gout was defined using only a questionnaire among NHANES 2007–2016 participants (n = 9,089). Hyperuricemia was observed in 1,798 (19.8%) adults, gout in 382 (4.2%). Of the 382 gout cases, 175 (1.9%) were accompanied by hyperuricemia and 207 (2.3%) were not. After correcting for age, sex, ethnicity, BMI, diabetes mellitus, hypertension, and eGFR, the association was significant at second (OR, 0.74; 95% CI, 0.55–0.99; $P$ = 0.049) and fourth quartiles (OR, 0.71; 95% CI, 0.52–0.94; $P$ = 0.029) of urinary molybdenum-to-creatinine ratio ($P$ = 0.037 for trend). When gout with hyperuricemia and gout without hyperuricemia were treated as two separate outcomes, urinary molybdenum-to-creatinine was associated with the former at the third (OR, 0.63; 95% CI, 0.41–0.98; $P$ = 0.041) and fourth quartiles (OR, 0.60; 95% CI, 0.38–0.94; $P$ = 0.025) ($P$ = 0.017 for trend), but not with the latter at any quartiles (Q2 OR, 0.85; 95% CI, 0.56–1.28; $P$ = 0.43; Q3 OR, 0.93; 95% CI, 0.62–1.39; $P$ = 0.73; Q4 OR, 0.86; 95% CI, 0.57–1.30; $P$ = 0.47; $P$ = 0.57 for trend).

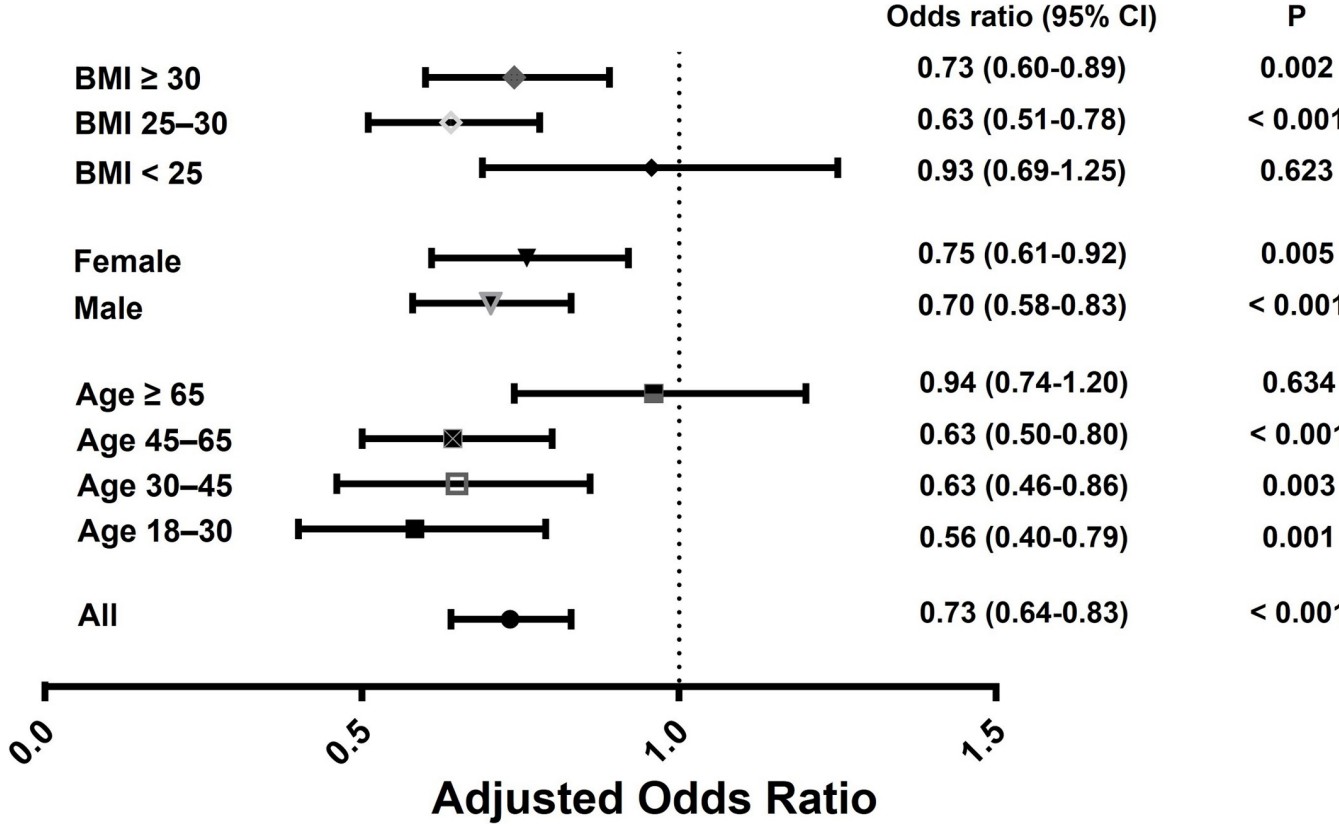

**Fig 2. Forest plot for the risk of hyperuricemia due to the increased urinary molybdenum levels.** The risk of hyperuricemia associated with urinary molybdenum levels (adjusted odds ratio of the 4th quartile vs. 1st quartile of urinary molybdenum) was compared according to age, sex, and BMI. Multivariate logistic regression analysis was adjusted for age, sex, ethnicity, BMI, diabetes mellitus, hypertension, and estimated glomerular filtration rate.

**Table 4. Association between urinary molybdenum levels and prevalence of gout.**

| | Gout | | Gout with hyperuricemia | | Gout without hyperuricemia | |
|---|---|---|---|---|---|---|
| | OR (95% CI) | *P* value | OR (95% CI) | *P* value | OR (95% CI) | *P* value |
| Urinary molybdenum-to-creatinine ratio | | 0.089 | | 0.067 | | 0.845 |
| | | 0.037[*] | | 0.017[*] | | 0.573[*] |
| Q1, reference | 1 | | 1 | | 1 | |
| Q2 | 0.74 (0.55, 0.99) | 0.049 | 0.76 (0.45, 1.04) | 0.076 | 0.85 (0.56, 1.28) | 0.427 |
| Q3 | 0.76 (0.56, 1.03) | 0.076 | 0.63 (0.41, 0.98) | 0.041 | 0.93 (0.62, 1.39) | 0.732 |
| Q4 | 0.71 (0.52, 0.94) | 0.029 | 0.60 (0.38, 0.94) | 0.025 | 0.86 (0.57, 1.30) | 0.472 |

Abbreviation: Q1–Q4, quartile group of urinary molybdenum levels.

[*]*P*-for-trend

Gout was defined using a questionnaire among NHANES 2007–2016 participants (n = 9,089).

Hyperuricemia is defined as a serum uric acid concentration of over 6.0 mg/dL for females and over 7.0 mg/dL for males.

Multivariate logistic regression analysis was adjusted for age, sex, ethnicity, BMI, diabetes mellitus, hypertension, and estimated glomerular filtration rate.

### Relationship between urinary molybdenum, systemic oxidative stress, and inflammation

In this study, we utilized serum gamma-glutamyltransferase levels as an indicator for systemic oxidative stress, and ferritin and C-reactive protein levels as systemic inflammation. Subjects with higher urinary molybdenum levels showed lower gamma-glutamyltransferase levels and lower ferritin and C-reactive protein levels (Table 1). In multivariable linear regression, urinary molybdenum levels were significantly associated with decreased gamma-glutamyltransferase and C-reactive protein levels in multivariable model (Table 5).

### Protective effects of molybdenum against $H_2O_2$-induced oxidative stress in HK-2 cells

Molybdenum was not significantly cytotoxic to HK-2 cells at doses below 200 μg/mL (Fig 3A). ROS concentration decreased when cells exposed to $H_2O_2$ were treated with molybdenum (Fig 3B); ROS decreased by 34% when given 1 μg/mL ($P < 0.01$), 30% when given 2 μg/mL ($P < 0.01$), 23% when given 5 μg/mL ($P < 0.01$), and 29% when given 10 μg/mL ($P < 0.001$) (n = 5 for each). Fig 3C shows the expression of antioxidant enzyme MnSOD decreased when cells were challenged with $H_2O_2$, but MnSOD expression increased dose-dependently with molybdenum treatment (n = 3 for each). These findings indicate molybdenum upregulated MnSOD expression and quenched oxidative stress in HK-2 cells.

### Discussion

We found molybdenum in human kidney proximal tubular epithelial cells provides antioxidative benefits in the presence of oxidative stress and that urinary molybdenum-to-creatinine ratio is associated with lower levels of systemic inflammation and oxidative stress and a reduced prevalence of hyperuricemia and gout in the general US adult population.

As our *in vitro* results indicate, molybdenum and oxidative stress have an interesting relationship. Agricultural studies have reported molybdenum-based fertilizers and supplements increase the expression of antioxidant enzymes in wheat and cabbage [25–27]. Molybdenum-based polyoxometalate nanoclusters, among other molybdenum-based nanoparticles [28, 29], are reported to provide antioxidative cryoprotection in human cells [30], suggesting that molybdenum's biochemical profile is beneficial for mitigating oxidative stress. Likewise, MnSOD upregulation and ROS reduction in our results demonstrate molybdenum increases tolerance against oxidative stress by upregulating antioxidative enzymes, perhaps directly scavenging ROS.

**Table 5. Association between urinary molybdenum levels[a] and systemic levels of oxidative stress and inflammation.**

| | Univariable analysis | | Multivariable analysis | |
|---|---|---|---|---|
| | β (95% CI) | *P* value | β (95% CI) | *P* value |
| GGT[a] | -0.070 (-0.086, -0.055) | < 0.001 | -0.052 (-0.067, -0.037) | < 0.001 |
| Ferritin[a] | -0.062 (-0.109, -0.016) | 0.008 | -0.020 (-0.061, 0.020) | 0.327 |
| C-reactive protein[a] | -0.117 (-0.156, -0.078) | < 0.001 | -0.184 (-0.220, -0.148) | < 0.001 |

Abbreviations: GGT, gamma-glutamyltransferase

[a]Urinary molybdenum-to-creatinine ratio (ng/mg) and outcome parameters were natural log-transformed.

Multivariable linear regression analyses were adjusted for age, sex, ethnicity, BMI, diabetes mellitus, hypertension, and estimated glomerular filtration rate.

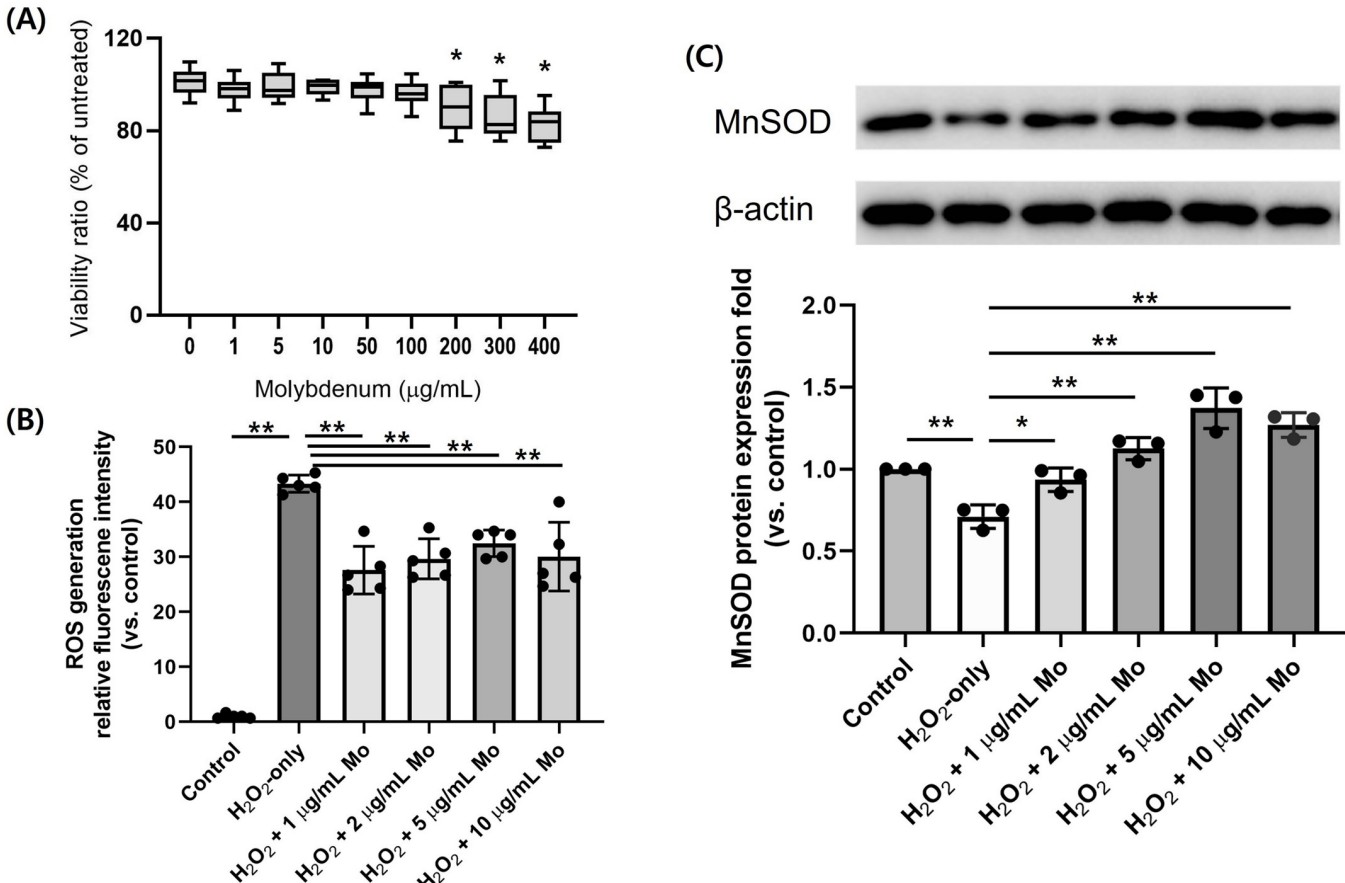

**Fig 3. Assessment of antioxidative capacity and ROS in HK-2 cells following molybdenum treatment and $H_2O_2$-induced oxidative stress.** (A) A viability assay was performed to determine the cytotoxic extent of molybdenum on HK-2 cells. Molybdenum significantly reduced cell viability at doses 200 μg/mL and above (n = 12 per dose). *$P < 0.05$ vs. untreated. (B) Oxidative stress was induced in HK-2 cells with 200 μM $H_2O_2$, and molybdenum was administered at 1, 2, 5, and 10 μg/mL. Molybdenum treatment significantly decreased cellular ROS at all doses (n = 5 per group). (C) Western blotting for manganese superoxide dismutase (MnSOD). Representative western blot image (upper panel) and measurement of western blot results (lower panel). The loading order of the western blot (upper panel) matches exactly with the order of the X-axis in the quantification results (lower panel). Molybdenum treatment increased the expression of cellular antioxidative MnSOD (n = 3 per group). Oxidative stress was induced in HK-2 cells with 100 μM $H_2O_2$, and molybdenum was administered at 1, 2, 5, and 10 μg/mL. Bar graph shows the mean and standard deviation levels. *$P < 0.05$, **$P < 0.01$; Mo, molybdenum; ROS, reactive oxygen species.

Main finding of this study is that increased urinary molybdenum levels are significantly associated with decreased uric acid levels or hyperuricemia risk. It is important to note that the concentration of molybdenum in blood (plasma or serum) or urine has different meanings. In a small study of four healthy young men who received dietary molybdenum for 24 days (molybdenum intake of 1,490 μg/day), plasma levels of molybdenum reached 6.22 ng/mL and average concentration of urinary molybdenum was 69 ng/mL [31]. Blood molybdenum concentrations peak at 1 hour after oral ingestion and then decline to baseline, and rapidly disappear within 1–2 hours after intravenous injection, with less than 50% detectable at 2 hours [32, 33]. Therefore, blood concentrations are considered as not suitable for the reflection of molybdenum status. Urine is the main excretion pathway for molybdenum [34, 35]. When molybdenum is injected intravenously, 34–60% of it is excreted in the urine after one day and 42–70% after five days [36]. About 60% of ingested molybdenum is eliminated in the urine at low intake levels (22 μg/day), but over 90% is excreted in the urine at high intake levels (467 μg/day or up to 1,488 μg/day) [34, 35]. Urinary molybdenum excretion is more directly connected to

recent dietary intake and seems to be a useful biomarker of short-term molybdenum intake, even if urinary molybdenum alone does not perfectly reflect molybdenum status [37].

Our cross-sectional analysis revealed a positive correlation between urinary molybdenum-to-creatinine ratios and higher eGFR. In general, a decrease in glomerular filtration decreases urinary excretion of solutes such as uric acid. Not all solute excretions are affected equally; solute size, hydrophilicity or lipophilicity, and charge all contribute to filtration and excretion. Therefore, when examining the association between urinary molybdenum and serum uric acid, including eGFR as a covariate was necessary. However, despite adjusting for eGFR, our regression models maintained urinary molybdenum was negatively correlated with serum uric acid levels and hyperuricemia.

This negative correlation between urinary molybdenum levels and serum uric acid concentration is a novel insight that contradicts previous studies. Several previous studies reported that molybdenum overexposure contributes to the development of hyperuricemia and gout, possibly via xanthine oxidase hyperactivity [10–12]. Nevertheless, our results indicate that the risk of hyperuricemia is reduced according to the increase in urinary molybdenum levels. These discrepancies might be derived from the difference in molybdenum intake amounts. In conditions where large amounts of molybdenum are ingested beyond the physiologic intake, such as in molybdenum intoxication, blood uric acid concentrations are increased by an intake that exceeds the increase in urinary uric acid excretion caused by molybdenum. Hyperuricemia was reported by Koval'skii et al. in patients with molybdenum poisoning; the patients' molybdenum intake was found to be 10 to 15 mg/day [11]. This greatly exceeds the molybdenum RDA of 45 μg/day and the tolerable UL of 2,000 μg/day. Seldén et al. reported that long-term occupational exposure to molybdenum increased serum uric acid levels and induced gout-like symptoms. In addition, avoidance of molybdenum exposure improved hyperuricemia and gout [38] (Occup Med (Lond). 2005 Mar;55(2):145–8. doi: 10.1093/occmed/kqi018). Deosthale et al. examined the association between molybdenum intake and uric acid levels in four volunteers and found no significant association at an intake amount in the 1.5 mg/day range [14]. Chappell et al. discovered a decrease in blood uric acid levels at a molybdenum intake of 7 μg/kg/day, which is near physiologic intake and is consistent with this study's findings [13].

Molybdenum's antioxidant effects may play a role in the relationship between increased urinary molybdenum and lowered risk of hyperuricemia. Oxidative stress is a known contributor to the development of CKD [39–42], a well-known risk factor for hyperuricemia and gout [43, 44]; emerging evidence proposes hyperuricemia and gout, in turn, are risk factors for CKD [45–48]. If molybdenum promotes xanthine oxidase activity, uric acid and free radical production would increase [49–51]. Nevertheless, if molybdenum's aforementioned antioxidant profile has a greater influence, it will provide renoprotection accompanied by increased uricosuria, ultimately decreasing serum uric acid content and lowering the risk of hyperuricemia. In addition, it is possible that molybdenum's anti-oxidative effect may have contributed to an increase in uric acid excretion through the intestine. Intestinal excretion of urate is important for the homeostasis of uric acid, and typically about one-third of the body's uric acid is excreted through the intestines [52]. Intestinal urate transporters are involved in this process, the most prominent of which is ATP-binding cassette subfamily G member 2 (ABCG2) [53]. ABCG2 is expressed in various tissues, including the kidney and intestine, and its dysfunction leads to hyperuricemia and gout [54]. A recent study has demonstrated that ROS inhibit ABCG2 through activation of ERK1/2 signaling [55]. Although uric acid excretion through the intestine is independent of kidney function, it is commonly affected by oxidative stress. The antioxidant effects of molybdenum may have prevented ABCG2 inhibition by ROS, thereby maintaining uric acid excretion and preventing the development of hyperuricemia.

This study confirmed that the decrease in serum uric acid concentration associated with urinary molybdenum occurred independently of the increase in eGFR. Recent studies indicate a link between uricosuria and antioxidants. Reports suggest antioxidant supplements, including vitamin C and E, reduce both oxidative stress and serum uric acid levels [56]. Choi et al. reported that taking 1.5 g of vitamin C daily is associated with a 45% reduced incidence of gout [57]; consuming 3.0 g/day resulted in transient increases in uric acid excretion in one study [58] and a 102% increase in uricosuria in another [59]. These findings that vitamin C promotes uricosuria have led to speculation that uric acid and vitamin C compete with each other for reabsorption [60]. Since both uric acid and vitamin C are reabsorbed at the proximal tubules [61, 62], increased reabsorption of vitamin C would interfere with uric acid reabsorption, resulting in greater uricosuria. A similar competitive relationship between the antioxidant molybdenum and uric acid for tubular reabsorption would have similar uricosuric results. While molybdenum reabsorption mechanisms in humans are not yet characterized [63], it should be noted that molybdenum is known to compete with sulfate for tubular reabsorption in ruminants [64].

Another important finding of this study was that increased urinary molybdenum was associated with a reduction in gout prevalence. Hyperuricemia is an important risk factor and pathophysiologic cause of gout development, so it is not surprising that molybdenum-induced reductions in uric acid and hyperuricemia were associated with a significant reduction in gout prevalence. However, gout can occur independently of kidney dysfunction or hyperuricemia [44]. Acute gouty arthritis is not accompanied by hyperuricemia in 12 to 43% of patients [65], as urate crystal formation can occur at normal or low uric acid concentrations due to factors such as changes in temperature, hydrophilicity, osmolarity, pH, or increased inflammatory response in the joint fluid [43]. More interestingly, this study demonstrated that urinary molybdenum was not associated with the gout without hyperuricemia. Therefore, the association between molybdenum and a decreased prevalence of gout in this study could be attributed to a urate-reducing effect of molybdenum.

Our cross-sectional analysis of the adult population is inherently unable to determine the causal relationship between molybdenum excretion and hyperuricemia. However, the analyses of antimony and tungsten as negative controls suggest that this particular relationship between molybdenum and hyperuricemia is not observed in many other metals (Table 3). It is possible that the effect of molybdenum in this study on uric acid levels was overestimated due to the large sample size. It also failed to account for factors that directly affect uric acid levels, such as dietary factors and nutrients. However, the strengths of this study are that it considered additional lifestyle-related factors such as socioeconomic factors, education, and smoking as modifiers, and analyzed various subgroups to prevent overestimation due to sample size and to provide multifaceted clinical evidence of the effects of molybdenum on uric acid. Our findings raise the need to investigate the cause and mechanism of this special relationship between molybdenum and hyperuricemia. Although we found the anti-oxidative effect of molybdenum both in the cohort study (association with decreased levels of GGT) and in cellular experiments (associated with decreased levels of ROS and increased levels of anti-oxidation enzyme levels), these findings are not direct evidence of the uric acid-lowering effects of molybdenum. To better understand molybdenum's biochemistry, additional *in vitro* work examining its role and antioxidant capacity could be performed in other cell types. Animal experiments or clinical studies to determine molybdenum's effects on urinary or intestinal uric acid excretion and urate transporters might be helpful. Follow-up epidemiological studies should also consider factors such as the population's dietary intake, including protein and antioxidants such as vitamin C.

To conclude, we demonstrated that a higher urinary molybdenum-to-creatinine ratio is associated with decreased serum uric acid and systemic inflammation, a relationship that

extends to lowered risks of hyperuricemia and gout. Using HK-2 cells, we identified the reno-protective potential of molybdenum via anti-oxidation mechanism against $H_2O_2$-induced oxidative stress. Altogether, the antioxidative effects of molybdenum may have contributed to the prevention of hyperuricemia observed epidemiologically. Future studies should confirm via *in vitro* and animal studies if molybdenum's antioxidative properties indeed directly increase the tubular excretion of uric acid and examine via human clinical trials whether molybdenum supplements decrease serum uric acid and help prevent hyperuricemia and gout.

## Supporting information

**S1 Table. Urinary molybdenum levels in relation to eGFR.**
(DOCX)

**S2 Table. Sensitivity analysis for the association between urinary molybdenum and prevalence of hyperuricemia.**
(DOCX)

**S3 Table. Subgroup analysis for the association between urinary molybdenum and prevalence of hyperuricemia according to sex.**
(DOCX)

**S4 Table. Subgroup analysis for the association between urinary molybdenum and prevalence of hyperuricemia according to age.**
(DOCX)

**S5 Table. Subgroup analysis for the association between urinary molybdenum and prevalence of hyperuricemia according to body mass index.**
(DOCX)

**S1 Method. Culture of HK-2 cells, cell viability assay, western blot analysis, and fluorescent microplate assay.**
(DOCX)

**S1 Raw images.**
(PDF)

**S1 Dataset.**
(CSV)

## Acknowledgments

**Institutional review board statement:** This cross-sectional observational cohort study was conducted according to the guidelines of the Declaration of Helsinki and approved by the Institutional Review Board of Seoul National University Boramae Medical Center (07-2019-35, 11 November 2019).

   **Informed consent statement:** Informed consent was initially obtained at the enrollment NHANES study and waived from all subjects during the analysis of this study.

## Author Contributions

**Conceptualization:** Kyungho Choi, Jung Pyo Lee, Jeonghwan Lee.

**Data curation:** Jeonghwan Lee.

**Formal analysis:** Joo Hong Joun, Lilin Li, Jung Nam An, Joonho Jang.

**Investigation:** Joo Hong Joun, Lilin Li, Joonho Jang.

**Resources:** Chun Soo Lim, Yon Su Kim, Kyungho Choi.

**Supervision:** Jung Pyo Lee, Jeonghwan Lee.

**Validation:** Lilin Li, Jeonghwan Lee.

**Writing – original draft:** Joo Hong Joun, Jeonghwan Lee.

**Writing – review & editing:** Yun Kyu Oh, Chun Soo Lim, Yon Su Kim, Kyungho Choi, Jung Pyo Lee.

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
