## [Decision Letter · Decision Letter 0]

8 Sep 2023

PONE-D-23-14643Antioxidative effects of molybdenum and its association with reduced prevalence of hyperuricemia in the adult populationPLOS ONE

Dear Dr. Lee,

Thank you for submitting your manuscript to PLOS ONE. After careful consideration, we feel that it has merit but does not fully meet PLOS ONE’s publication criteria as it currently stands. Therefore, we invite you to submit a revised version of the manuscript that addresses the points raised during the review process.

We look forward to receiving your revised manuscript.

Kind regards,

Donovan Anthony McGrowder, PhD., MA., MSc

Academic Editor

PLOS ONE

Journal Requirements:

"Jeonghwan Lee received funding of this study. This work was supported by the Seoul Metropolitan Government Seoul National University Boramae Medical Center (clinical research grant-in-aid 04-2020-3)."

7. Your ethics statement should only appear in the Methods section of your manuscript. If your ethics statement is written in any section besides the Methods, please delete it from any other section. 

**Additional Editor Comments:**

Dear Dr. Lee,

 Your manuscript “Antioxidative effects of molybdenum and its association with reduced prevalence of hyperuricemia in the adult population” has been assessed by our reviewers. They have raised a number of points which we believe would improve the manuscript and may allow a revised version to be published in PLOS ONE. Their reports, together with any other comments, are below.

If you are able to fully address these points, we would encourage you to submit a revised manuscript to PLOS ONE by the date given below.

Best regards,

Dr. Donovan McGrowder

Reviewers' comments:

Reviewer's Responses to Questions

**Comments to the Author**

1. Is the manuscript technically sound, and do the data support the conclusions?

Reviewer #1: Yes

Reviewer #2: Yes

Reviewer #3: Yes

2. Has the statistical analysis been performed appropriately and rigorously? 

Reviewer #1: Yes

Reviewer #2: Yes

Reviewer #3: No

3. Have the authors made all data underlying the findings in their manuscript fully available?

Reviewer #1: Yes

Reviewer #2: Yes

Reviewer #3: Yes

4. Is the manuscript presented in an intelligible fashion and written in standard English?

Reviewer #1: Yes

Reviewer #2: Yes

Reviewer #3: Yes

5. Review Comments to the Author

Reviewer #1: The manuscript is well-written and scientifically sound but the topic is moderately novel and not of much priority for publications at PLosONE. Therefore, I see that the manuscript of low priority for acceptance.

Reviewer #2: Review

The research evaluated the epidemiological connection between urinary molybdenum levels and hyperuricemia, as well as kidney-related outcomes, in 15,370 adult participants from the National Health and Nutrition Examination Survey (NHANES) conducted between 1999 and 2016. Antimony and tungsten were used as control trace metals.

In the adult NHANES population, the urinary molybdenum-to-creatinine ratio was significantly associated with reduced serum uric acid levels, a lower prevalence of hyperuricemia, and a decreased likelihood of gout. Higher urinary molybdenum levels were linked to lower levels of systemic oxidative stress (measured by gamma glutamyl transferase levels) and inflammation (C-reactive protein levels). In HK-2 cells subjected to H2O2-induced oxidative stress, molybdenum upregulated the expression of manganese superoxide dismutase and reduced oxidative stress.

In summary, urinary molybdenum levels are associated with a decreased prevalence of hyperuricemia and gout in the adult population. Molybdenum's antioxidative properties appear to play a crucial role in reducing systemic inflammation, reactive oxygen species, and uric acid levels.

Minor comment

1. Do the authors offer an explanation for the discrepancies in their findings when compared to previous studies?

Reviewer #3: In this manuscript “Antioxidative effects of molybdenum and its association with reduced prevalence of hyperuricemia in the adult population”, Dr. Lee and colleagues carried out the large population-based cohort studies about the relationship between urinary molybdenum concentration and urinal acid metabolism. The authors showed higher urinal molybdenum relates to lower prevalence of hyperuricemia and gout. The authors also indicated that molybdenum decreased oxidative stress in HK-2 cell line and suggested molybdenum’s antioxidant effect will provide renoprotection accompanied by increased uricosuria.

I believe that this research will have a major impact on elucidating the physiological activity of molybdenum. Therefore, I recommend the authors considering the following points.

1. Methods part: The authors should provide a sample size calculation. When numerous cases (actually over 1500 cases) are included in the statistics, analysis power is substantially increased. This implies an exaggerated tendency to reject null hypotheses with clinically negligible differences. Table 1 shows significant differences in many clinical factors through the multivariate analysis. However, some of these significant differences are too small to discuss at the clinical level (e.g. serum albumin levels are 4.3 ± 0.4 and 4.2 ± 0.4 between lower and upper half of urinal molybdenum concentration). Additionally, I suspect that there are some potential confounding factors in table 1 parameters. For example, the amount of serum creatinine is said to be significantly lower in the upper half of urinal molybdenum group than that of the lower half. However, at the same time, the upper half of urinal molybdenum group also has a significantly higher proportion of females, who have lower serum creatinine level than males. I am afraid that the multivariate analysis could not correct for the effects of confounding factors because the excessive number of cases.

2. Results part: Please indicate mean of urinal molybdenum concentration and urinary molybdenum-to-creatinine ratio (including upper and lower half group) clearly in the result part of main manuscript. Not in the supplemental section.

3. Table 3: Q1and Q4, which is higher urinal molybdenum group? Please indicate all mean values of metal concentrations in each quartile.

4. Discussion part: Please discuss the relationship between molybdenum concentration of blood and that of urine. Some articles and / or pilot studies referring to the relationship should be provided.

5. Discussion part: Acute gout attack without hyperuricemia is frequently observed (11%-49%). The authors indicated urinary molybdenum-to-creatinine ratio is not correlated with gout unaccompanied by hyperuricemia. Please discuss about the reasons why protecting effect of molybdenum was not detected in the gout patients who had normal serum uric acid level.

6. Discussion part: Current studies have shown that the intestine is an important potential organ for the excretion of uric acid outside the kidneys (e.g. A. Hosomi et al. Extra-renal elimination of uric acid via intestinal efflux transporter BCRP/ABCG2, PLoS One. 7 (2012) e30456). Please discuss about the possible influences of ingested and circulating molybdenum on uric acid metabolism in the intestinal tract.

6. PLOS authors have the option to publish the peer review history of their article (what does this mean?). If published, this will include your full peer review and any attached files.

Reviewer #1: **Yes: **Doaa Attia

Reviewer #2: No

Reviewer #3: **Yes: **Kinya Okamoto

---

## [Author Response · Author response to Decision Letter 0]

16 Nov 2023

Reviewer #1: 

The reviewers pointed out the limitations and strengths of this manuscript in detail. Some of the physiological roles of molybdenum and experimental evidence have been reported previously and are discussed in the manuscript. However, relatively little is known about the biological role of molybdenum in a physiological condition in the human body. This paper enrolled a relatively large number of participants in this study and showed that urinary excretion of molybdenum is related to the prevention of hyperuricemia and gout. The results of this study, together with various statistical analyses and experimental results, can help expand our understanding of molybdenum.

Reviewer #2: 

: Thank you for important comment. As mentioned in the Introduction section, some studies report that molybdenum is associated with increased uric acid levels and increased incidence of gout, and some studies report no association. However, this study reverses the trend and reports that increased molybdenum is associated with lower uric acid levels and gout incidence.

These discrepancies might be derived from the difference in molybdenum intake amounts. In conditions where large amounts of molybdenum are ingested beyond the physiologic intake, such as in molybdenum intoxication, blood uric acid concentrations are increased by an intake that exceeds the increase in urinary uric acid excretion caused by molybdenum. Hyperuricemia was reported by Koval’skii et al. in patients with molybdenum poisoning; the patients' molybdenum intake was found to be 10 to 15 mg/day [11]. This greatly exceeds the molybdenum Recommended Dietary Allowance (RDA) of 45 μg/day and the Tolerable Upper Intake Level (UL) of 2,000 μg/day. Deosthale et al. examined the association between molybdenum intake and uric acid levels in four volunteers and found no significant association at an intake amount in the 1.5 mg/day range [14]. Chappell et al. discovered a decrease in blood uric acid levels at a molybdenum intake of 7 g/kg/day, which is near physiologic intake and is consistent with the authors' findings [38]. 

 Following the reviewer’s valuable comments, we added this description on discrepancy in the Discussion section (line 295-307). 

[11] Koval'skii VV, Iarovaia GA, Shmavonian DM Modification of human and animal purine metabolism in conditions of various molybdenum bio-geochemical areas. Zh Obshch Biol. 1961;22:179-91. 

[14] Deosthale YG, Gopalan C The effect of molybdenum levels in sorghum (Sorghum vulgare Pers.) on uric acid and copper excretion in man. Br J Nutr. 1974 31(3):351-5. 10.1079/bjn19740043

[38] Chappell WR, Meglen RR, Moure-Eraso R, Solomons CC, Tsongas TA, Walravens PA, Winston PW. 1979. Human Health Effects of Molybdenum in Drinking Water. EPA-600/1-79-006. Cincinnati, OH: U.S. Environmental Protection Agency, Health Effects Research 

Reviewer #3: 

3-1: Thank you for the valuable comments. The reviewer pointed out two critical points related to the fact that the number of subjects included in the study was too large to determine the true effect. The first is that a large number of subjects may overinterpret clinically unmeaningful effects, and the second is that multivariable adjustment may not be sufficient due to the presence of confounding effects between variables. The authors totally agree with these concerns.

 Following the comments, the authors calculated an adequate sample size. Various studies have provided statistical calculations for the appropriate sample size in logistic regression (Hsieh et al. Stat Med. 1998;17(14):1623; Bull et al. Am J Epidemiol. 1993;137(6):676; Demidenko et al. Stat Med. 2007;26(18):3385; Kim et al. Stat Methods Med Res. 2017;26(3):1237).

- OR (odd ratio) 1.10 

- Alpha = 0.05 

The significance level (α, alpha) represents the probability of rejecting the null hypothesis when it is true, leading to a type I error.

- Beta = 0.01–0.20 (Power = 1 – beta = 0.80–0.99)

The test power (1 – β) is the probability of rejecting the null hypothesis if false, effectively controlling for type II error (β)

- Calculated total N = from 3,458 to 8,092 

Calculating sample size in clinical studies is typically aimed at avoiding situations where there are too few enrolled participants to show the statistical significance of the actual effect. Our study enrolled more subjects than the simple calculation; however, there is a risk that the large number of participants may exaggerate the true effect. To control the confounding effects of variables and reduce the risk of overestimation, the authors added the following analyses: First, additional multivariable logistic regression models were tested by adding significant covariates, including NHANES survey cycles, serum albumin levels, socioeconomic status, education status, and smoking.

We tested three multivariable models.

- Model 1: covariates of age, sex, ethnicity, BMI, diabetes mellitus, hypertension, and estimated glomerular filtration rate.

- Model 2: model 1 covariates + NHANES survey cycle (1999-2000 1st, … 2015-2016 9th)

- Model 3: model 2 covariates + socioeconomic status, education levels, and smoking. 

Urinary molybdenum levels were significantly associated with hyperuricemia in all three models.

 Supplementary Table 2. Sensitivity analysis for the association between urinary molybdenum and prevalence of hyperuricemia

The second solution is subgroup analysis. Subgroup analysis allowed us to limit the number of participants in the group to the appropriate number of patients, which we identified to be between 3,000 and 4,000 through statistical calculations, and to determine whether the study's key findings were more or less significant in any of the subgroups. We analyzed participants by gender, age group, and BMI. The effect of uric acid reduction by molybdenum in urine was more pronounced in men, younger participants, and overweight participants, and less significant in elderly patients over 65 years of age and normal weight participants.

These reanalysis results and discussions are described in the Results and Discussion section as follows: 1. (Result section, line 213-219) “In the sensitivity analysis, multivariable logistic regression models were further adjusted using covariates of socioeconomic status, education levels, and smoking status (Supplementary Table 2). The effects of urinary molybdenum to hyperuricemia were stratified according to sex, age group, and BMI criteria (Supplementary Table 3-5). Uric acid lowering effect of molybdenum more pronounced in men, younger participants, and overweight participants, and less significant in elderly patients over 65 years old and normal weight participants.”, 2. (Discussion section, line 357-363) “It is possible that the effect of molybdenum in this study on uric acid levels was overestimated due to the large sample size. It also failed to account for factors that directly affect uric acid levels, such as dietary factors and nutrients. However, the strengths of this study are that it considered additional lifestyle-related factors such as socioeconomic factors, education, and smoking as modifiers, and analyzed various subgroups to prevent overestimation due to sample size and to provide multifaceted clinical evidence of the effects of molybdenum on uric acid.”

3-2. 

: Thank you for the valuable comment. The description of the urinary concentration of molybdenum and the ratio of urinary molybdenum to creatinine concentration is fundamentally important to all. We have added them to Table 1 and mentioned them in the main text (line 172-176) as follows: “The median and interquartile range for urinary molybdenum were 43.4 (22.7–73.7) ng/mL, and those for the molybdenum-to-creatinine ratio were 39.5 (27.0–58.6) ng/mg. Participants with higher urinary molybdenum-to-creatine levels had more than twice the urinary molybdenum-to-creatine ratio [58.6 (47.6, 79.1) vs. 27.0 (20.0, 33.3); P < 0.001].”

3-3

: Thank you for the important comment. Q4 is the highest group of urinary metal concentrations. Following the comment, we added the detailed values of median and interquartile ranges in the Table 3 legend as follows: “Urinary molybdenum-to-creatinine ratio, median (IQR), ng/mg, Q1 20.0 (15.1, 23.7), Q2 33.3 (30.3, 36.4), Q3 47.6 (43.4, 52.7), Q4 79.1 (66.5, 106,4); Urinary antimony-to-creatinine ratio, median (IQR), ng/mg, Q1 0.03 (0.02, 0.04), Q2 0.05 (0.05, 0.06), Q3 0.08 (0.07, 0.09), Q4 0.15 (0.13, 0.22); Urinary tungsten-to-creatinine ratio, median (IQR), ng/mg, Q1 0.03 (0.02, 0.03), Q2 0.05 (0.05, 0.06), Q3 0.09 (0.08, 0.10), Q4 0.19 (0.14, 0.28)”. 

3-4

: Thank you for the important comment. The detailed discussions on the relationship between molybdenum concentration of blood and urine are added in the discussion section (line, 268-283) as follows: “It is important to note that the concentration of molybdenum in blood (plasma or serum) or urine has different meanings. In a small study of four healthy young men who received dietary molybdenum for 24 days (molybdenum intake of 1,490 μg/day), plasma levels of molybdenum reached 6.22 ng/mL and average concentration of urinary molybdenum was 69 ng/mL [31]. Blood molybdenum concentrations peak at 1 hour after oral ingestion and then decline to baseline, and rapidly disappear within 1-2 hours after intravenous injection, with less than 50% detectable at 2 hours [32,33]. Therefore, blood concentrations are considered as not suitable for the reflection of molybdenum status. Urine is the main excretion pathway for molybdenum [34,35]. When molybdenum is injected intravenously, 34-60% of it is excreted in the urine after one day and 42-70% after five days [36]. About 60% of ingested molybdenum is eliminated in the urine at low intake levels (22 μg/day), but over 90% is excreted in the urine at high intake levels (467 μg/day or up to 1,488 μg/day) [34,35]. Urinary molybdenum excretion is more directly connected to recent dietary intake and seems to be a useful biomarker of short-term molybdenum intake, even if urinary molybdenum alone does not perfectly reflect molybdenum status [37].”

3-5

: Thank you for the important comment. The detailed discussions on the reasons why protecting effect of molybdenum was not detected in the gout patients without hyperuricemia are added in the discussion section (line, 342-353) as follows: “Another important finding of this study was that increased urinary molybdenum was associated with a reduction in gout prevalence. Hyperuricemia is an important risk factor and pathophysiologic cause of gout development, so it is not surprising that molybdenum-induced reductions in uric acid and hyperuricemia were associated with a significant reduction in gout prevalence. However, gout can occur independently of kidney dysfunction or hyperuricemia [65]. Acute gouty arthritis is not accompanied by hyperuricemia in 12 to 43% of patients [66], as urate crystal formation can occur at normal or low uric acid concentrations due to factors such as changes in temperature, hydrophilicity, osmolarity, pH, or increased inflammatory response in the joint fluid [67]. More interestingly, this study demonstrated that urinary molybdenum was not associated with the gout without hyperuricemia. Therefore, the association between molybdenum and a decreased prevalence of gout in this study could be attributed to a urate-reducing effect of molybdenum.”

3-6

: Thank you for the valuable comment. The detailed discussions on the possible influences of ingested and circulating molybdenum on uric acid metabolism in the intestinal tract are added in the discussion section (line, 316-326) as follows: “In addition, it is possible that molybdenum’s anti-oxidative effect may have contributed to an increase in uric acid excretion through the intestine. Intestinal excretion of urate is important for the homeostasis of uric acid, and typically about one-third of the body's uric acid is excreted through the intestines [52]. Intestinal urate transporters are involved in this process, the most prominent of which is ATP-binding cassette subfamily G member 2 (ABCG2) [53]. ABCG2 is expressed in various tissues, including the kidney and intestine, and its dysfunction leads to hyperuricemia and gout [54]. A recent study has demonstrated that ROS inhibit ABCG2 through activation of ERK1/2 signaling [55]. Although uric acid excretion through the intestine is independent of kidney function, it is commonly affected by oxidative stress. The antioxidant effects of molybdenum may have prevented ABCG2 inhibition by ROS, thereby maintaining uric acid excretion and preventing the development of hyperuricemia.”

---

## [Decision Letter · Decision Letter 1]

29 Jan 2024

PONE-D-23-14643R1Antioxidative effects of molybdenum and its association with reduced prevalence of hyperuricemia in the adult populationPLOS ONE

Dear Dr. Lee,

Thank you for submitting your manuscript to PLOS ONE. After careful consideration, we feel that it has merit but does not fully meet PLOS ONE’s publication criteria as it currently stands. Therefore, we invite you to submit a revised version of the manuscript that addresses the points raised during the review process.

Your manuscript was reviewed by a new reviewer as original reviewer was not available. Few  minor comments were provided. Please address all comments as appropriate.

We look forward to receiving your revised manuscript.

Kind regards,

Partha Mukhopadhyay, Ph.D.

Section Editor

PLOS ONE

Journal Requirements:

Reviewers' comments:

Reviewer's Responses to Questions

**Comments to the Author**

1. If the authors have adequately addressed your comments raised in a previous round of review and you feel that this manuscript is now acceptable for publication, you may indicate that here to bypass the “Comments to the Author” section, enter your conflict of interest statement in the “Confidential to Editor” section, and submit your "Accept" recommendation.

Reviewer #3: All comments have been addressed

Reviewer #4: (No Response)

2. Is the manuscript technically sound, and do the data support the conclusions?

Reviewer #3: Yes

Reviewer #4: Yes

3. Has the statistical analysis been performed appropriately and rigorously? 

Reviewer #3: Yes

Reviewer #4: Yes

4. Have the authors made all data underlying the findings in their manuscript fully available?

Reviewer #3: Yes

Reviewer #4: Yes

5. Is the manuscript presented in an intelligible fashion and written in standard English?

Reviewer #3: Yes

Reviewer #4: Yes

6. Review Comments to the Author

Reviewer #3: I think that the authors responded to the first reviewer’s comments in good faith.

I cannot find further revising points.

Reviewer #4: I have pointed out a few minor issues to addressed before acceptance of the manuscript which seemed relevant to me to match upto the standards of the journal.

7. PLOS authors have the option to publish the peer review history of their article (what does this mean?). If published, this will include your full peer review and any attached files.

Reviewer #3: **Yes: **Kinya Okamoto

Reviewer #4: No

---

## [Author Response · Author response to Decision Letter 1]

3 May 2024

4-1. Thank you for valuable comment. In this study, we did not show the direct mechanism of the uric acid lowering effects of molybdenum. This is the study's main limitation. However, we found novel findings of a decreased prevalence of hyperuricemia according to the increase in urinary molybdenum levels. We performed thorough statistical analysis, including sensitivity analysis and subgroup analysis, with other metal (antimony and tungsten) controls. In addition, we suggested the antioxidative effect of molybdenum in both the NHANES cohort study (associated with decreased levels of GGT, an index of systemic oxidative stress) and in vitro experiments. In the discussion section, we added the limitations and strengths of this study as follows: “Our cross-sectional analysis of the adult population is inherently unable to determine the causal relationship between molybdenum excretion and hyperuricemia.” , “Although we found the anti-oxidative effect of molybdenum both in the cohort study (association with decreased levels of GGT) and in cellular experiments (associated with decreased levels of ROS and increased levels of anti-oxidation enzyme levels), these findings are not direct evidence of the uric acid-lowering effects of molybdenum.” , “Animal experiments or clinical studies to determine molybdenum’s effects on urinary or intestinal uric acid excretion and urate transporters might be helpful.”

4-2. Thank you for the valuable comments. The strength of this study is that increased molybdenum intake is associated with lower uric acid levels and a lower frequency of hyperuricemia in the physiologic condition of the general population with normal molybdenum intake. As mentioned in the Introduction and Discussion section, there is a known risk of hyperuricemia or gout at high doses of molybdenum (10 to 15 mg/day) that exceed the usual intake [Recommended Dietary Allowance (RDA) is 45 μg/day or Tolerable Upper Intake Level (UL) is 2,000 μg/day] [11,12]. However, there have also been reports that no increased risk of hyperuricemia or gout has been observed at standard doses [13,14]. In contrast to these previous studies, our results show that increased intake or exposure, expressed as an increase in urinary molybdenum, is associated with a decrease in serum uric acid levels and a decrease in the prevalence of gout. 

 We suggest that the differences in molybdenum intake may explain why our results differ from other previous studies. During this second revision, we could find additional papers that linked high intakes of molybdenum to increased uric acid and gout. Selde´n et al. reported that long-term occupational exposure to molybdenum increased serum uric acid levels and induced gout-like symptoms. In addition, avoidance of molybdenum exposure improved hyperuricemia and gout (Occup Med (Lond). 2005 Mar;55(2):145-8. doi: 10.1093/occmed/kqi018). 

 In the Discussion section, we added the related articles and reasonable explanations as follows: “This negative correlation between urinary molybdenum levels and serum uric acid concentration is a novel insight that contradicts previous studies. Several previous studies reported that molybdenum overexposure contributes to the development of hyperuricemia and gout, possibly via xanthine oxidase hyperactivity [10-12]. Nevertheless, our results indicate that the risk of hyperuricemia is reduced according to the increase in urinary molybdenum levels. These discrepancies might be derived from the difference in molybdenum intake amounts. In conditions where large amounts of molybdenum are ingested beyond the physiologic intake, such as in molybdenum intoxication, blood uric acid concentrations are increased by an intake that exceeds the increase in urinary uric acid excretion caused by molybdenum. Hyperuricemia was reported by Koval’skii et al. in patients with molybdenum poisoning; the patients' molybdenum intake was found to be 10 to 15 mg/day [11]. This greatly exceeds the molybdenum RDA of 45 μg/day and the tolerable UL of 2,000 μg/day. Selde´n et al. reported that long-term occupational exposure to molybdenum increased serum uric acid levels and induced gout-like symptoms. In addition, avoidance of molybdenum exposure improved hyperuricemia and gout (Occup Med (Lond). 2005 Mar;55(2):145-8. doi: 10.1093/occmed/kqi018). Deosthale et al. examined the association between molybdenum intake and uric acid levels in four volunteers and found no significant association at an intake amount in the 1.5 mg/day range [14]. Chappell et al. discovered a decrease in blood uric acid levels at a molybdenum intake of 7 μg/kg/day, which is near physiologic intake and is consistent with this study’s findings [38].

4-3. Thank you for the valuable comment. In response to the reviewer's request for rigorous statistical tests during the initial revision, we conducted sensitivity analyses and subgroup analyses based on age category, sex, and BMI category. These additional statistical analyses provided further insight on the substantial connection between urine molybdenum levels and hyperuricemia. In this second revision, we selected forest plots as a more intuitive method to display these findings and provided them as the main Figure 3.(Refer to Figure 3)

Figure 3. Forest plot for the risk of hyperuricemia due to the increased urinary molybdenum levels. The risk of hyperuricemia associated with urinary molybdenum levels (adjusted odds ratio of the 4th quartile vs. 1st quartile of urinary molybdenum) was compared according to age, sex, and BMI. Multivariate logistic regression analysis was adjusted for age, sex, ethnicity, BMI, diabetes mellitus, hypertension, and estimated glomerular filtration rate. 

4-4. Thank you for the valuable comment. In general, Kaplan-Meier analysis is suitable for analyzing outcomes with event time. However, this study reports novel findings of a significant association between urinary metal (molybdenum) concentrations and serum uric acid levels from the NHANES cohort study with a cross-sectional design. In the absence of information on the time of hyperuricemia’s onset, we performed logistic regression analysis (instead of Cox regression or Kaplan-Meir analysis) and the results was visualized using the spline curve in Figure 1. In the current dataset, Kaplan-Meier or Cox regression is impractical due to the lack of information on the onset time of hyperuricemia (cross-sectional analysis design).

In Figure 2a, the results of kidney tubular cell viability according to molybdenum concentration are also presented, but there are no differences in the event time (all viability data was measured 24 hours after molybdenum exposure), so we believe that the current presentation method is appropriate. 

4-5. Thank you for the important comment. The loading order of the western blot in Figure 2C (upper panel) matches exactly with the order of the x-axis in the quantification results for the western blot shown below (lower panel). All of the raw data for the western blot is included in the Supplements, and the loading order matches X-axis labeling of Figure 2-C. To clarify the results description, the authors corrected Figure 2-C’s figure legend as follows: The authors corrected Figure 2-C’s figure legend as follows: (C) Western blotting for manganese superoxide dismutase (MnSOD). Representative western blot image (upper panel) and measurement of western blot results (lower panel). The loading order of the western blot (upper panel) matches exactly with the order of the X-axis in the quantification results (lower panel). Molybdenum treatment increased the expression of cellular antioxidative MnSOD (n = 3 per group). Oxidative stress was induced in HK-2 cells with 100 μM H2O2, and molybdenum was administered at 1, 2, 5, and 10 μg/mL. Bar graph shows the mean and standard deviation levels. *P < 0.05, **P < 0.01; Mo, molybdenum; ROS, reactive oxygen species

 The figures in the supplementary data also include the western band loading sequences.

---

## [Decision Letter · Decision Letter 2]

10 Jun 2024

Antioxidative effects of molybdenum and its association with reduced prevalence of hyperuricemia in the adult population

PONE-D-23-14643R2

Dear Dr. Lee,

We’re pleased to inform you that your manuscript has been judged scientifically suitable for publication and will be formally accepted for publication once it meets all outstanding technical requirements.

Kind regards,

Partha Mukhopadhyay, Ph.D.

Section Editor

PLOS ONE

Additional Editor Comments (optional):

Reviewers' comments:

Reviewer's Responses to Questions

**Comments to the Author**

1. If the authors have adequately addressed your comments raised in a previous round of review and you feel that this manuscript is now acceptable for publication, you may indicate that here to bypass the “Comments to the Author” section, enter your conflict of interest statement in the “Confidential to Editor” section, and submit your "Accept" recommendation.

Reviewer #4: All comments have been addressed

2. Is the manuscript technically sound, and do the data support the conclusions?

Reviewer #4: Yes

3. Has the statistical analysis been performed appropriately and rigorously? 

Reviewer #4: Yes

4. Have the authors made all data underlying the findings in their manuscript fully available?

Reviewer #4: Yes

5. Is the manuscript presented in an intelligible fashion and written in standard English?

Reviewer #4: Yes

6. Review Comments to the Author

Reviewer #4: The comments has been addressed. The justifications have made the manuscript more decipherable. The manuscript looks fine to me to approve acceptance now.

7. PLOS authors have the option to publish the peer review history of their article (what does this mean?). If published, this will include your full peer review and any attached files.

Reviewer #4: No

---

## [Editor Report · Acceptance letter]

27 Jun 2024

PONE-D-23-14643R2 

PLOS ONE

Dear Dr. Lee, 

I'm pleased to inform you that your manuscript has been deemed suitable for publication in PLOS ONE. Congratulations! Your manuscript is now being handed over to our production team.

Kind regards, 

on behalf of

Dr. Partha Mukhopadhyay 

Section Editor

PLOS ONE